# The Adverse Outcome Pathway Framework Applied to Neurological Symptoms of COVID-19

**DOI:** 10.3390/cells11213411

**Published:** 2022-10-28

**Authors:** Helena T. Hogberg, Ann Lam, Elan Ohayon, Muhammad Ali Shahbaz, Laure-Alix Clerbaux, Anna Bal-Price, Sandra Coecke, Rachel Concha, Francesca De Bernardi, Eizleayne Edrosa, Alan J. Hargreaves, Katja M. Kanninen, Amalia Munoz, Francesca Pistollato, Surat Saravanan, Natàlia Garcia-Reyero, Clemens Wittwehr, Magdalini Sachana

**Affiliations:** 1National Toxicology Program Interagency Center for the Evaluation of Alternative Toxicological Methods, National Institute of Environmental Health Sciences, National Institutes of Health, Research Triangle Park, NC 27518, USA; 2Johns Hopkins University, Baltimore, MD 21205, USA; 3Green Neuroscience Laboratory, Neurolinx Research Institute, San Diego, CA 92111, USA; 4Physicians Committee for Responsible Medicine, Washington, DC 20016, USA; 5Institute for Green & Open Sciences, Toronto, ON M6J 2J4, Canada; 6A.I. Virtanen Institute for Molecular Sciences, University of Eastern Finland, 70211 Kuopio, Finland; 7European Commission, Joint Research Centre (JRC), 21027 Ispra, Italy; 8Division of Otorhinolaryngology, Department of Biotechnologies and Life Sciences, University of Insubria, Ospedale di Circolo e Fondazione Macchi, 21100 Varese, Italy; 9School of Science and Technology, Nottingham Trent University, Nottingham NG11 8NS, UK; 10European Commission, Joint Research Centre (JRC), 2440 Geel, Belgium; 11Centre for Predictive Human Model Systems Atal Incubation Centre-Centre for Cellular and Molecular Biology, Hyderabad 500039, India; 12Environmental Laboratory, US Army Engineer Research & Development Center, Vicksburg, MS 39180, USA; 13Environment Health and Safety Division, Environment Directorate, Organisation for Economic Cooperation and Development (OECD), 75016 Paris, France

**Keywords:** AOP, SARS-CoV-2, neuropathology, anosmia, encephalitis, stroke, human-specific research

## Abstract

Several reports have shown that the severe acute respiratory syndrome coronavirus 2 (SARS-CoV-2) has the potential to also be neurotropic. However, the mechanisms by which SARS-CoV-2 induces neurologic injury, including neurological and/or psychological symptoms, remain unclear. In this review, the available knowledge on the neurobiological mechanisms underlying COVID-19 was organized using the AOP framework. Four AOPs leading to neurological adverse outcomes (AO), anosmia, encephalitis, stroke, and seizure, were developed. Biological key events (KEs) identified to induce these AOs included binding to ACE2, blood–brain barrier (BBB) disruption, hypoxia, neuroinflammation, and oxidative stress. The modularity of AOPs allows the construction of AOP networks to visualize core pathways and recognize neuroinflammation and BBB disruption as shared mechanisms. Furthermore, the impact on the neurological AOPs of COVID-19 by modulating and multiscale factors such as age, psychological stress, nutrition, poverty, and food insecurity was discussed. Organizing the existing knowledge along an AOP framework can represent a valuable tool to understand disease mechanisms and identify data gaps and potentially contribute to treatment, and prevention. This AOP-aligned approach also facilitates synergy between experts from different backgrounds, while the fast-evolving and disruptive nature of COVID-19 emphasizes the need for interdisciplinarity and cross-community research.

## 1. Introduction

Since the beginning of the COVID-19 pandemic, scientific reports have described cases of neurologic injury [1,2] and explored the potential neurotropic nature of severe acute respiratory syndrome coronavirus 2 (SARS-CoV-2) [3]. The neurological impact has become an increasingly important area of study as global data collection has shown this to be one of the most common, persistent, and debilitating effects of COVID-19 (e.g., Long-COVID) [4,5,6]. Here, we consider evidence using the Adverse Outcome Pathways (AOP) framework [7] as well as new multiscale pathway perspectives with the aim of sharing the advantages, limitations, challenges, and potential evolution of utilizing the AOP approach for studying the neurological symptoms induced by SARS-CoV-2.

The link between SARS-CoV-2 and specific neurological effects became apparent from reports suggesting the onset of anosmia and ageusia as early diagnostic markers [8]. As the COVID-19 pandemic progressed, additional findings on long-term neurological complications after recovery began to emerge [9], including long-term olfactory disturbance (i.e., hyposmia and parosmia) accompanied by cognitive and emotional dysregulation that could lead to dementia [10]. A broad range of neurological complications has since been reported in patients with SARS-CoV-2, such as ischemic or hemorrhagic stroke, encephalitis, Guillain-Barré syndrome, brain fog, headaches, dizziness, fatigue, nausea, myalgia, sleep disorders, confusion, and seizures [11,12,13]. However, the mechanism by which SARS-CoV-2 induces neurologic injury remains unclear and requires further study by systematic human-based in vitro and long-term epidemiological studies. In that context, a working group focused on the neurological-related mechanisms of COVID-19 was formed within the CIAO project [14] “Modelling the Pathogenesis of COVID-19 using the AOP framework” [15,16]. The charge of this working group was to organize the evidence on the neuropathological mechanisms underlying COVID-19 using the AOP framework [7].

The AOP framework emerged from a need of the chemical regulatory and scientific community to better understand and communicate the complex mechanisms taking place in response to chemicals leading to human and environmental health adverse outcomes (AO). Since 2012, a program dedicated to AOP development and application operates under the auspices of the Organisation for Economic Cooperation and Development (OECD) [17]. Breaking down complex biological pathways into linear constructs and reassembling them into AOP networks can provide mechanistic understanding based on the available biological and toxicological evidence [18,19,20]. Due to its fundamental conceptual principles and guidelines, it is beginning to be applied also within the biomedical field [21].

Causally connected biological key events (KE) that are essential for the manifestation of an AO and that are triggered by a stressor in a molecular initiating event (MIE) can be captured within an AOP [22,23]. The causality information is stored in key event relationships (KERs). Information related to modulating factors can also be accommodated in KERs, allowing for a better understanding of the variation in disease progression and susceptibility in different individuals. The online repository of the AOPs is the AOP-knowledgebase [24], which includes the main AOP developing platform, named AOP-Wiki [25]. Importantly, AOPs are living documents that can be continuously updated as new knowledge emerges. The interdisciplinary nature of the AOPs is another benefit and the AOP-Wiki allows scientists from diverse fields to work together using the same framework and online repository platform. Thus, an AOP-aligned approach facilitates synergy between experts while the fast-evolving and disruptive nature of COVID-19 emphasizes the need for interdisciplinarity and cross-community research.

However, the pathogenic potential, the variability in the clinical symptoms of COVID-19, and the fast-changing variants make the application of the AOP framework challenging in this context. Mutations have led to COVID-19 waves of infection with significantly varying symptoms, prognoses, and transmissibility levels. Reports to date are mainly associated with coronavirus structural proteins and, in particular, the receptor binding domains of spike (S)-proteins rather than other non-structural and accessory proteins [26]. For example, SARS-CoV-2 B.1.617.2, also termed the Delta variant, was classified as a variant of concern according to the World Health Organization (WHO), demonstrating both increased transmissibility and increased disease severity. In contrast, although the Omicron (B.1.1.529) variant seemed more infectious than the Delta variant and more likely to have vaccine breakthroughs [27], it has been reported to induce milder symptoms in most patients. The majority of the work presented here took into account the initially published literature related to the globally dominant variant named D614G, which showed higher transmissibility, but no increased disease severity compared to its ancestral strain. Beyond variant properties, research has also shown that SARS-CoV-2 viral loads can significantly influence the disease intensity and progression [28], parameters that need to be accommodated for in KERs descriptions.

Here, we integrate the currently available knowledge on neurological manifestations linked to SARS-CoV-2 using the classical AOP approach. The most recent version of the AOPs presented here can be found in the AOP-Wiki [25]. Modulating and multiscale factors are also discussed in the present paper. 

In parallel with building AOPs depicting neurological outcomes in COVID-19, the working group discussed whether classical AOPs—considering molecular mechanisms as the initiating events of the disease-can fully capture and elucidate the mechanisms of COVID-19, in general, and the neuro-psychosocial aspects, in particular. Incorporating a multiscale perspective that can capture the causal nature of the pandemic across scales was investigated, and the benefits from a new spatio-temporal and recursive approach were explored [29]. Such a shift would be important for the mitigation of the current outcome at the individual and population levels, as well as future pandemic prevention. Although consensus was not always achieved in the mapping of the putative AOPs or the scale of approach, the structure of the neuro working group allowed for open discussion, in which scientific primary and metadata were collected, interpreted, and added to increase the understanding of the COVID-19 neurological symptoms. This publication is an outcome of the CIAO project, where experts work together to assemble AOPs and evolve the framework in ways that can help elucidate the mechanisms underlying COVID-19 pathophysiology [29].

## 2. COVID-19-Related Neuro Events and Adverse Outcome Pathways

### 2.1. ACE2 Receptor Interaction

A classical AOP generally starts with an MIE. In the present case, SARS-CoV-2 was identified as the initiating stressor. Several studies have shown that SARS-CoV-2 enters cells by binding the surface S-protein to the angiotensin-converting enzyme 2 (ACE2) cell-surface receptor (Event 1739 in AOP Wiki) [30,31,32,33]. After the initial binding, transmembrane protease serine 2 (TMPRSS2), which is anchored to the outer surface of the host cell membrane, has been shown to cleave the S-protein at a specific site, facilitating a conformational change in the latter that promotes virus-cell membrane fusion and subsequent viral entry into the host cell [31,34]. The ACE2 receptor and TMPRSS2 are expressed in a wide variety of tissues, including regions of the brain. A recent study identified the expression levels of ACE2 in both endothelial cells and non-vascular cells, composed predominantly of neurons but also with low expressions of astrocytes and microglia [35]. The expression varied among the brain regions with high levels in the pons, visual cortex, and amygdala, and reduced levels in the midbrain, cerebellar cortex, dentate nucleus, and medulla. However, it is important to note that not all the cells of the nervous system possess the machinery that is necessary for the interaction with SARS-CoV-2 [36], and it is possible that the virus may infect certain cells of neuronal origin through a different set of extracellular host receptors, such as neuropilin-1 [37]. In this respect, neuropilin-1 has been detected in olfactory epithelium ensheathing cells and there is some evidence showing that SARS-CoV-2 can enter the brain through the olfactory nerve [38,39]. It has also been suggested that SARS-CoV-2 S-protein can produce an inflammatory response in brain endothelial cells, resulting in the disruption of the blood–brain barrier (BBB) integrity and enabling the virus to enter the brain [40]. Furthermore, it has been shown that the S-protein can impair the vascular and immune regulatory functions of brain pericytes, which could explain vascular-mediated brain damage [41]. However, to date, there has been limited detection of viral proteins in the brain and cerebrospinal fluid (CSF) [42,43], indicating that the neurological effects may be secondary due to damage to endothelial cells and the trafficking of cytokines into the brain from the blood [44]. Concordantly, recent publications rule out the possibility that the axons of olfactory sensory neurons (OSNs) constitute a virus gateway to the central nervous system (CNS) [45,46], but the transsynaptic transfer of SARS-CoV-2 from peripheral neuron infection remains a possibility. Consequently, the MIEs and earlier KEs leading to AOs in the nervous system need to be further evaluated and confirmed.

From a multiscale perspective, it is interesting to consider that the focus should not only be on ACE2 as an initiating event in AOPs. Including environmental, social, and individual scales could help to mitigate the spread and prevent future pandemics. In addition, the precise SARS-CoV-2 molecular pathways might not be fully understood and may be altered depending on the target cell type and variants. Recent research has identified important non-ACE2 factors in excessive inflammation involving key monocytes and macrophages lacking ACE2 receptors pathways [47].

### 2.2. Blood–Brain Barrier (BBB) Disruption

The BBB is crucial in protecting the hemodynamic function of the brain. In addition, it is a key structural defense against toxic substances and microorganisms such as viruses and chemicals. The complex structure of the BBB, which comprises a number of cell types and junctions, mechanically and biochemically regulates the permeability of xenobiotics [48]. It is, therefore, not surprising to see the clinical reports of neuroinflammatory and BBB disruption markers described repeatedly in the literature. However, it remains an open question as to how and when these processes may occur. The interconnected nature of brain capillary endothelial cells (BCECs), pericytes, neurons, astrocytes, and microglia in the BBB, strongly suggest this to be a path of SARS-CoV-2 viral entry to the brain, and a contribution to neuroinflammatory events [49]. A compromised BBB is one of the proposed KEs in the network of neuro AOPs (Figure 1) linked to several AOs such as strokes, seizures, anosmia, and encephalitis [50]. Evidence from in vitro models showed that isolated spike proteins can cross the BBB [40,51]. Viral particles were also identified in the frontal lobe by Paniz-Mondolfi and colleagues, suggesting a hematological route of infection through the BBB [52]. ACE2 expression has been reported in the BBB [42,51], and although in low abundance, its blockage with anti-ACE2 antibody in human induced pluripotent stem cells (hiPSC) led to the almost complete blockage of infection [51]. Elevated expressions of TMPRSS2 and NRP1 were reported in an hiPS-derived BCEC model, and their role in SARS-CoV-2 infection was demonstrated with the use of specific antibodies and inhibitors [51]. Alterations of the interferon-γ-mediated signaling pathways in COVID-19 patients and in a cellular model [51] revealed another potential explanation for the compromised BBB, specifically through the disruption of the brain endothelial cells [53] resulting in increased proinflammatory cytokines within an individual even without viral entry into the brain.

### 2.3. Hypoxia and Hypoxemia

Hypoxia and hypoxemia are KEs involved in severe neurological complications seen in COVID-19. Brain injury from hypoxia and hypoxemia are commonly reported in fatal COVID-19. Two separate studies showed that all enrolled patients who died had acute hypoxic damage, without evidence of thrombi or vasculitis, throughout many brain structures after infection with SARS-CoV-2 [54,55]. Frontera and colleagues identified hypoxemia, along with sepsis and uremia to be among the most common causes of multifactorial toxic-metabolic encephalopathy (TME) [56]. Of these factors, hypoxia was the risk factor most associated with TME, thus supporting the hypothesis that hypoxia is one of the crucial key events [57,58]. However, hypoxia is also identified as a potential mechanism in stroke (see below). In Figure 1, hypoxia is identified as a KE in lung, stroke, and seizure AOPs; however, the KER is still under development.

### 2.4. Neuroinflammation and Oxidative Stress

Neuroinflammation is a crucial event in the onset and consolidation of SARS-CoV-2-related neurological sequelae and neurodegenerative illnesses [59,60]. Neuroinflammation can be detected by the presence of activated microglia and astrocytes. Patients with moderate to severe COVID-19 are often characterized by the presence of elevated plasma levels of glial fibrillary acidic protein (GFAP), which is a biochemical indicator of astrocytic activation [61]. Changes in glial cell morphology, increased antigen expression, and the increased proliferation of the glial cells in the affected brain regions represent classical hallmarks of neuroinflammation [62,63,64,65,66,67]. 

Both astrocytes and microglia modulate the activation of inflammatory signaling pathways. After this, there is increased expression and/or release of inflammatory cytokines’, eicosanoids’, and metalloproteinases’ [68] production of reactive oxygen (ROS) and nitrogen species (RNS) [69]. Microglia and astrocytes vary in their activation and there is an interplay between pro-inflammatory, anti-inflammatory signaling, and cellular functions, including phagocytosis [65,70]. Rather than a direct migration of the virus into the brain through the nasal cavity and the olfactory pathway, or penetration of the virus across the BBB, immune activation, the entering of activated immune cells due to disrupted BBB, and inflammation within the brain are currently considered as the most plausible triggers of neurologic disease occurring as a consequence of SARS-CoV-2 infection in acute COVID-19 [44]. As part of the BBB, astrocytes can receive circulating pro-inflammatory cytokine signals, and/or activate microglia leading to neuroinflammation [71]. Patients affected by SARS-CoV-2 can present with a large increase in systemic pro-inflammatory cytokines and the induction of reactive pro-inflammatory microglia, which can upregulate the expression of neuroinflammation-related genes [57,68,69]. Analysis of cerebrospinal fluid (CSF) samples from COVID-19 patients showed the upregulation of interferon-regulated gene expression in dendritic cells, the activation of T cells and NK cells, as well as elevated levels of IL1 and IL12 [72]. This immune response seems to be compartmentalized, as suggested by the presence of clonal expansion of T cells and antibodies specific for SARS-CoV-2 spike protein in the CSF [73]. Additionally, specific markers of monocyte activation and neuronal injury were also found in the CSF during this acute phase of the disease [74].

### 2.5. Neurodegeneration

Several pathological outcomes have been proposed and are under investigation to determine how SARS-CoV-2-induced damage is associated with neurodegeneration. SARS-CoV-2-induced systemic inflammation, or the cytokine storm, may cause disruption of the BBB, neural and glial cell damage or dysfunction, and penetration of pro-inflammatory cytokines into the CNS. These pro-inflammatory events could for example alter the ability of microglial cells to phagocytose amyloid beta, promoting the accumulation of amyloid plaques, a known hallmark of Alzheimer’s disease (AD) [75]. In addition, systemic inflammation could cause the activation of neural-immune cells for the further induction of pro-inflammatory cytokine production in the brain. Subsequently, these cytokines could contribute to synaptic dysfunction and neuronal loss and lead to neurodegeneration associated with, for example, AD, Parkinson’s disease, and multiple sclerosis [76].

SARS-CoV-2 induces hypoxic alterations and demyelinating lesions in the brains of some individuals [51,75,76]. The outcomes of these alterations have been shown in follow-up studies of recovered SARS-CoV-2 patients, indicating alterations in brain functional integrity, specifically in the hippocampus. The hippocampus is the memory house of the brain, and its atrophy has been linked to cognitive decline [4,77]. Furthermore, severely infected individuals have characteristic hypercoagulation and disseminated intravascular coagulation [78], which may lead to reduced perfusion of the white matter, potentially leading to ischemic white matter damage. SARS-CoV-2-induced hypoperfusion in the brain can also increase the phosphorylation of tau, the main component of neurofibrillary tangles found in AD-affected brains. This is a characteristic feature of the early stage of AD and cognitive decline [79,80]. Furthermore, it has been suggested that SARS-CoV-2-induced neurotropism may lead to an increased risk of AD development and the progression of AD-like symptoms [81]. SARS-CoV-2 infection in the CNS can lead to neuroinflammation, which activates downstream signaling, including the increased release of pro-inflammatory markers, intense oxidative stress, and ineffective innate immune responses [82,83,84]. In the presence of certain risk factors or comorbidities, neuroinflammation can become prolonged or uncontrollable and can lead to an increased risk of neurodegeneration and disease [85]. Moreover, SARS-CoV-2-infected individuals with comorbidities that display existing inflammation, for example, diabetes, atherosclerosis, or sub-clinical dementia, could potentially be more at risk of neurodegenerative disorders.

### 2.6. Anosmia

Olfactory disturbances were the most usual neurological symptoms of COVID-19 for the initial variants of SARS-CoV-2. Loss of smell (anosmia, if complete loss of smell) has been included as a common symptom of COVID-19 [86]. The prevalence, intensity, and duration of these olfactory symptoms vary from patient to patient and are affected by epigenetic and geographic modulating factors [87,88]. The causes of the differences are not known and are driven by variability in the human population but also by the variation of SARS-CoV-2 variants that are currently in circulation. The self-rated and objective smell evaluation by patients also contributes to a different prevalence, the scoring being more subjective.

The exact mechanism of anosmia is still unknown, but there are probably several factors that contribute to the loss of smell. Developing an AOP based on current evidence from the literature enabled to propose underlying mechanisms and to identify current knowledge gaps guiding further research [89]. SARS-CoV-2 initiates with the binding of the viral spike proteins to the ACE2 receptors. In the olfactory epithelium, ACE2 protein is located mainly in the sustentacular cells but is not expressed in the olfactory receptor neurons (ORN) as demonstrated by ACE2 immunohistochemical expression [90]. The damage to the sustentacular cells leads to the subsequent damage of the ORN with the complete or partial loss of the normal sense of smell. The rapid regeneration of sustentacular cells due to stem cell maturation correlates to the recovery of the sense of smell that is clinically observed in most COVID-19 cases. The time course of smell recovery in COVID-19 is around one week [39] but can also last substantially longer in some individuals (long-term anosmia). Measuring olfactory function revealed changes in smell perception that could last up to 15 months after symptom onset. In these cases, alterations at the CNS level, showed also in a reorganization of ORN nuclear architecture and a widespread downregulation of olfactory receptors [91], could be associated with neuroinflammation leading to neurodegeneration-inducing persistent symptoms [92].

The mechanisms leading to short-term anosmia involving the olfactory neuroepithelium have been detailed in AOP394, entitled “SARS-CoV-2 infection of olfactory epithelium leading to impaired olfactory function (short-term anosmia)” which can be found in the AOP-Wiki [93]. This AOP describes the current mechanistic understanding of the causative links between the binding of spike protein to ACE2 receptors on sustentacular cells (MIE), which leads to sustentacular cell depletion, olfactory sensory neuron decreases, and olfactory epithelium degeneration (KEs), resulting in anosmia (AO) (Figure 1) [94].

### 2.7. Encephalitis

It has been recently suggested that a transient viral infection of the brain may occur in the early phase of infection, and/or that viral antigens may be present in the brain at low concentrations [44]. Although infrequently detected, infected CNS cells do not present with clusters of surrounding inflammatory cells, which suggests that the presence of the virus in the CNS may not stimulate classic viral encephalitis [44].

However, some studies have reported the presence of SARS-CoV-2 in the CNS and CSF of patients with acute neurologic symptoms specific to encephalitis [3,95]. Encephalitis has been clinically observed in COVID-19 patients. It is characterized by acute onset, and the symptoms include headache, fever, vomiting, convulsions, and consciousness disorders [3,96]. In the ongoing pneumonia epidemic, viral encephalitis was confirmed by the presence of SARS-CoV-2 in the CSF of patients with COVID-19 [97]. Moreover, the presence of SARS-CoV-2 viral particles in the endothelial cells and the pericytes of brain capillaries and neurons [49,97,98], as well as in the glial cells (microglia and astrocytes) [98] was observed in the post-mortem examinations of SARS-CoV-2-infected patients.

The binding of spike protein to ACE2 receptors on endothelial cells, pericytes of brain capillaries, as well as microglia and astrocytes, triggers their activation, which eventually results in the formation and release of proinflammatory cytokines/chemokines, nitric oxide, prostaglandin E2, ROS, and RNS. As a consequence, these proinflammatory factors can trigger neuronal cell death by well-known mechanisms [99,100,101] contributing, together with brain neuroinflammation, to encephalitis.

Currently, an AOP entitled “Binding of SARS-CoV-2 spike protein to ACE 2 receptors expressed on brain cells (neuronal and non-neuronal) leads to neuroinflammation resulting in encephalitis” is under development [102]; this AOP describes the current mechanistic understanding of the causative links between the binding of spike protein to ACE2 receptors on brain cells (MIE), which leads to glia activation causing neuroinflammation and neurodegeneration (KE), resulting in encephalitis (AO) (Figure 1).

### 2.8. Stroke

Large vessel strokes have been reported in a number of cases in patients diagnosed with COVID-19 [85,103,104]. As cerebrovascular disease is not a common clinical finding in patients with SARS-CoV-2, many predisposition factors such as cardiovascular comorbidities, including hypertension (HTN) and diabetes mellitus (DM), were proposed to play a role. However, age might not be a critical risk factor as there are reports showing younger patients with occlusions of large vessels linked to SARS-CoV-2 infection [103]. Since the beginning of the pandemic, patients diagnosed with stroke attributed to SARS-CoV-2 infection were also found to have elevated disseminated intravascular coagulation (DIC) and increased numbers of venous thromboembolisms [104].

The exact mechanism of COVID-19-induced stroke is not well established, although evidence has emerged indicating that SARS-CoV-2 can directly cause hypercoagulopathy, arteritis, and vascular endothelial dysfunction, which can lead to ischemic stroke [105]. Hypoxia and the excessive secretion of inflammatory cytokines have also been proposed as potential mechanisms involved in large vessel strokes in COVID-19 complications [85,106]. However, there are several reports hypothesizing that cerebrovascular disease in COVID-19 patients might not be due to the direct viral mechanism but rather the result of other conditions that are present in patients with severe COVID-19 during hospitalization (e.g., heart failure, septic shock, coagulopathy, and acute cardiac injury), which are well established factors potentially predisposing patients to stroke [103,104,107]. SARS-CoV-2 is known to block ACE2 receptors, which are regulators of blood pressure. This can lead to the functional underexpression of ACE2, thereby increasing the risk of hemorrhagic stroke. In individuals with hypertension, who already experience decreased expressions of ACE2 and difficulties with controlling blood pressure, SARS-CoV-2 infection may be more likely to create neurological complications characterized as cerebrovascular hemorrhage [103,106].

Markers that have been associated with cerebrovascular disease in patients positive for SARS-CoV-2 include elevated d-dimer or fibrin degradation product levels, reduced platelet counts, and transient increases in serum inflammatory cytokines and antiphospholipid antibodies [85,104,107]. The access of SARS-CoV-2 to cerebral vasculature is thought to be achieved through the general circulation, most likely by breaching the BBB and affecting the parenchyma [85].

An AOP entitled “Binding of SARS-CoV-2 spike protein to ACE 2 receptors expressed on pericytes leads to disseminated intravascular coagulation resulting in cerebrovascular disease (stroke)” is under development in the AOP Wiki [107]; this AOP describes the current mechanistic understanding of the causative links between the binding of spike protein to ACE2 receptors on pericytes (MIE), which leads to BBB disruption, DIC, and thrombosis (KEs), and results in cerebrovascular disease (AO) (Figure 1).

### 2.9. Seizure and Epilepsy 

The AOP on seizure and epilepsy linked to COVID-19 is still under development but not yet in the AOP Wiki. Seizures are defined as uncontrolled electrical activity disturbances in the brain and can lead to both motor and behavioral symptoms [108]. Experiencing more than two seizures within 24 h is typical of epilepsy. In larger COVID-19 data sets, epilepsy is reported in 0.5–4% of patients [13,109,110]. Potential KEs that can lead to seizure in these patients includes hypoxia [109], neuroinflammation [110], and a compromised BBB [111,112,113] (Figure 1). A recent systematic review identified 62 manuscripts that reported on patients with COVID-19 and seizures [114]. Many of the papers were case studies describing patients with new onsets of focal seizures, serial seizures, and status epilepticus. The interpretation by the authors in this study was that organ failure, metabolic derangements, drug–drug interactions, or brain damage were potential causes of seizures. Though very rare, there have been case studies describing other SARS-CoV infections of the brain linked to seizures [115,116], indicating that other KEs may be involved in this AOP.

### 2.10. Integration of Classical AOPs

The modularity of AOPs allows the self-assembling and construction of AOP networks. Through an AOP network, shared KEs and KERs are emerging, and knowledge gaps are being identified [22,117]. Furthermore, the assembling of AOP networks provides an opportunity to better visualize the commonalities and core pathways and mechanisms (Figure 1). Through the AOP network, the viral spike protein binding to ACE2 located in endothelial, neuronal, and glial cells is presented as an important step for long-term anosmia compared to short-term anosmia, which involves binding to ACE2 found in the sustentacular and basal cells of the olfactory epithelium. Neuroinflammation is emerging as a common KE for many individual AOPs that describe the disease process that can lead to long-term anosmia, encephalitis, stroke, and seizures. 

The recognition of the role of low brain oxygen also helps to elucidate the connection of neurological symptoms to the well-studied pulmonary distress identified with COVID-19. Hypoxemia and hypoxia serve as illustrations of how multiple systems, in this case, pulmonary and neuronal, can recursively interact and lead to AOs (Figure 1). Moreover, it is an example of how mechanisms ranging from the molecular to higher biological levels in COVID-19 pathways can be integrated to better understand the disease.

Within the CIAO project, there is a parallel effort to create and publish a more comprehensive AOP network that covers all the possible mechanisms and AOs induced by SARS-CoV-2 beyond the nervous system. This assembling of neuro-related AOPs into a network as described in Figure 1 was the outcome of the neuro working group discussion and is open to revision as new data become available and as potential new KEs and/or pathways emerge.

## 3. Additional Factors Impacting Neurological Effects of COVID-19

### 3.1. Modulating Factors

Since the beginning of the pandemic, the breadth of symptoms linked to SARS-CoV-2 infection has puzzled the medical and scientific communities. Many intrinsic or extrinsic factors seem to modulate the presence and severity of symptoms. Modulating factors could be defined as variables known to alter the shape of the response–response function that describes the quantitative relationship between two KEs [118]. Those factors should be explored and incorporated into the AOP development when enough information is available. Age, sex, genetic susceptibility or resistance, co-morbidities (obesity, historic dyslipidemia, pre-existing heart failure), vitamin D deficiency, diet, and environmental factors such as air pollution and exposure to chemicals have been investigated as factors modulating COVID-19 underlying mechanisms based on the current knowledge available in the literature [119]. Pre-existing cardiovascular disease, as well as hypertension, diabetes mellitus, and obesity, have been linked to increased severity of neurological symptoms [120]. Another study analyzed the long-term effects using a prospective online survey and identified older age, female sex, and disease severity as risk factors for persistent neuropsychiatric symptoms [121]. Although many of these studies are not specifically related to neurological symptoms, the connection between these systems and the brain is well established. Longitudinal studies will thus be essential to help improve our understanding of the linkage between modulating factors and neurological outcomes during and after COVID-19 [44]. Age as a modulating factor is of special interest to the neurological symptoms and is discussed here.

#### 3.1.1. Pediatric and Fetal Exposure

Even though SARS-CoV-2 infections have been shown to induce milder symptoms in children, reports are showing increased complications also in younger patients, especially with the newer strains of the virus [122]. Many of the more severe cases are describing neurological symptoms such as encephalitis, seizure, and Guillain-Barré syndrome. Though it is unclear if this effect is due to the virus entering the brain or a secondary effect due, for example, to general inflammation, it is of concern as it is well-described that the developing brain is particularly susceptible to perturbations. Even though the most vulnerable window of brain development takes place during the fetal stage and the first 5 years of life, the brain is considered to continue developing and maturing during adolescence [123]. 

There is limited knowledge about the impact of SARS-CoV-2 on the very immature brain in fetuses and infants. In the few studies of pregnant women with SARS-CoV-2, most have reported negative results for the presence of the virus in the neonates and placenta [124]. Two studies have observed placental invasion by immunohistochemical analysis and electron microscopy [125,126], and there are case reports which detected the virus in one infant and one fetus suffering from COVID-19 [127,128]. The concern for harmful effects on the developing brain due to SARS-CoV-2 remains, as it is well-documented that maternal infection is a risk factor for neurodevelopmental disorders including autism spectrum disorder and schizophrenia [129,130,131,132]. The main proposed key events are associated with the increased cytokine storm and hyperinflammation observed in pregnant women with COVID-19 and the loss of functional placenta integrity. However, there are several other potential KEs and modulating factors that might play a role such as prolonged fever, hypoxia, hypertension, and the effects of medication. One major challenge to understanding the impact of SARS-CoV-2 on the developing brain is the delayed observation of neurodevelopmental AOs, as many of these will not be observed for another few years. This suggests that the closer monitoring of babies born to women that tested positive for SARS-CoV-2 during pregnancy would be of value. 

#### 3.1.2. Exposure in the Elderly

Elderly COVID-19 patients have been shown to have higher risks of neurological complications [133]. A cross-hospital clinical study in the UK examined neurological and psychiatric measures and revealed that 62% of the patients had cerebrovascular events. In total, 74% of the patients had ischemic stroke, 23% developed unspecified encephalopathy, and 1% acquired CNS vasculitis [134]. Of the patients with cerebrovascular events, 82% were older than 60 years. As people age, it is more common to have comorbid conditions that can influence the severity of the SARS-CoV-2 infection [135]. Furthermore, a systematic review effort associated the manifestation of neurological symptoms in elderly COVID-19-infected patients with higher disease severity and mortality rates [136], which were further supported by a more recent rapid review approach [136].

AD is among the most common CNS-associated comorbidities of COVID-19 [137,138], and the relationship or interaction between the two is complex. AD and related dementias (ADRD) are some of the most prevalent neurodegenerative disorders. These diseases are characterized by the deposition of amyloid plaques and neurofibrillary tangles in the brain regions responsible for memory and learning, causing dementia. Dementia patients, including those with AD, are increasingly susceptible to SARS-CoV-2 infection severity and mortality [139,140] possibly due to the increased expression of ACE2 observed in AD patients [141]. Similar increases in the ACE2 gene have been observed in a genome-wide association study using the brain tissues of AD patients [142]. Another possible factor contributing to the connection between COVID-19 and AD is genetic predisposition. A recent study indicates that the e4 allele of apolipoprotein E4 (APOE4), which is a major genetic risk factor for AD, also increases susceptibility to SARS-CoV-2 when compared to individuals with the e3 allele of APOE4 [143,144]. The increased risk is associated with a lower level of antiviral gene expression in APOE4 e4 individuals as compared to those with APOE4 e3 [145]. In another gene, OAS1, was found to be linked to both AD and COVID-19 [146], yet further studies are needed to fully address the details of the interaction. However, various similarities, differences, and interactions between COVID-19 and ADRD have been proposed [147,148]. 

### 3.2. Multiscale Perspective

Notwithstanding correlations between various factors and COVID-19, there is a deeper question of causality that goes beyond modulation. For example, neuro-psychological components should not just be viewed simply as modulating or outcome components but also as critical factors that may precede infection and may determine outcomes at the individual and population levels. 

A multiscale pathway perspective was explored to chart the full range of spatiotemporal scales and factors to consider, understand, and respond to COVID-19 and future pandemics [119]. These factors and their representations, such as disparities and psychological stress were then ported to the neuro-domain work.

The number of disparities and preexisting health conditions that might impact the evolution of COVID-19 is extensive and beyond the scope of this paper. However, shown here are examples that illustrate how the centrality of such factors at the individual and population levels are identified. There is a discussion of how these factors can be considered under the AOP framework and an acknowledgment that neuropsychological factors may be present and critical at the beginning of the causal chain and trigger the neurological events and outcomes (Figure 2). These may be as many risk factors as an effect. By extension, neurological factors may also play a causal role in terms of understanding recovery and Long-COVID [149].

#### 3.2.1. Psychological Stress

Psychological stress has been associated with adverse health outcomes [150], especially neurological disorders as it can exacerbate neuroinflammatory processes (Figure 2) [151,152,153]. The proposed mechanism of this psychosocial determinant of health is, in part, due to priming microglia and inflammatory responses, which contribute to cognitive deterioration [154]. Similarly, SARS-CoV-2 causes a hyperinflammatory response with the activation of monocytes, dendritic cells, mast cells, T cells, and endothelial cells [53]. Vicarious traumatization, self-quarantine, social distancing, anxiety induced by public awareness of the disease, pre-existing medical conditions, and feelings of worry about potential exposure, serve as psychological stressors that may worsen the prognosis of individuals infected with SARS-CoV-2 [155,156,157,158]. Stress has had a profound effect on family members, workers, and communities affected by the suffering or loss of life brought about by the pandemic. A recent study found that family members had significant symptoms of posttraumatic stress disorder (PTSD) months after the admission of a loved one to an intensive care unit and that these neuropsychological symptoms were significantly associated with ethnicity and gender [159]. Thus, considering psychosocial stress could help elucidate how disparities can contribute to the neuropathogenesis of COVID-19 and to outcomes in both patients and their communities.

#### 3.2.2. Disparities

Studies of long-standing systemic health and social inequities have shown that some groups are at higher risk of illness from COVID-19 [160]. For example, food insecurity has been associated with a higher likelihood of hospitalization in COVID-19 patients [161]. Conversely, a global study in low- and middle-income countries showed that knowing someone infected with COVID-19 increased the probability of experiencing food insecurity [162]. According to data-tracking studies and surveys performed by the Census Bureau and the Center on Budget and Policy Priorities, millions of people in the U.S. are suffering from food insecurity, which affects more households with children, households of people of color, and people working in low-paid industries [163]. The pandemic significantly increased the food insecurity further in the U.S., especially for low-income Americans [164]. There are several studies showing the impact of nutrition and lifestyle on the immune system [165] and how appropriate nutrition can contribute to the prevention, management, and recovery from COVID-19, including neurological symptoms (Figure 2) [166,167]. Additional factors related to disparities such as the inability to secure other resources that are important during this pandemic (e.g., face masks, sanitizing products, adequate health care services, and informational resources) have also been major factors in the prevention and treatment of COVID-19.

Given the bidirectional relationship between socioeconomic status and long-term health outcomes, research focusing on the social determinants of health in the research of COVID-19 and in human health, in general, is needed. The inclusion of these initiation events that expand beyond the classical AOP framework may provide more adaptive and predictive outcomes, as described elsewhere in the CIAO project [29].

## 4. Discussion

As the pandemic progressed and evidence accumulated, it became evident that SARS-CoV-2 was also neurotropic [168], putting in focus the nervous system in clinical management and research. However, these findings do not imply that the nervous system is only directly affected by SARS-CoV-2. Indirect effects on the central and peripheral nervous system through injury to other organ systems are also reported [169]. Some underlying conditions and risk factors have been linked to the severity of the reported neuropathological conditions that may partially explain the selectivity in certain patients infected by SARS-CoV-2. Moreover, certain societal factors might modulate the effects of SARS-CoV-2 infection by preceding spread and altering outcomes. Thus, multiscale considerations are important for prevention and mitigation. A One Health approach, recognizing the interconnection between people, animals, and the environment, can help to improve the global response to COVID-19 [170].

Several challenges with developing AOPs for COVID-19 were discussed in this article, and there has been an effort to develop a three-dimensional AOP by incorporating modulating factors as actual KEs. In the current AOP framework, modulating factors are documented and described in the KER, but within COVID-19, they might play a bigger role that requires different visualization. Several risk factors are still unknown; for example, the mutations of the virus over time seem to influence AOs that might impact the developed AOPs as well. In addition, poverty, nutrition, and psychological stress challenge the molecular approach in the initiation of the disease and might contribute to explaining some of the differences noted in the severity of the disease between individuals and populations. 

Another challenge is the massive number of published articles, often without peer review, which makes it difficult to select good-quality papers. The aim of the CIAO project is to make sense of this overwhelming flood of publications by reviewing and organizing the knowledge into AOPs. In addition, within the CIAO project, a parallel effort led by the evidence-based toxicology collaboration [171] aims to perform a systematic scoping review to identify and map KEs from the available literature that describes neurological effects due to SARS-CoV-2. The literature is being organized based on KEs and AOs and will support KERs in developed AOPs. Such an approach can also contribute to the identification of new KEs, AOs, and data gaps in an unbiased manner. The AOPs presented in the present paper are expected to evolve and reach maturation and will potentially increase the branching of the AOP network in Figure 1 once the literature review from the systematic scoping effort has been completed.

As evidence is starting to become available linking cognitive decline and anosmia to brain pathologies such as reductions in grey matter thickness and the overall brain size in SARS-CoV-2 patients [172], it seems that is increasingly important to focus efforts on linking these structural changes to the AOPs presented in this paper. Studies on brain tissue damage due to SARS-CoV-2 infection were lacking at the time that the work presented here was taking place, but the accumulated knowledge in this field is expected to identify additional KEs related to anatomical abnormalities and potentially bring to light risk factors related to lifestyle, environmental parameters, and preexisting health issues and how these factors potentially can impact the progression of neurological symptoms. Information about COVID-19 pathogenesis will continue to appear and neurological impacts will continue to be reported, notably linked to Long-COVID [173,174]. In addition, the disease might contribute to neurodevelopmental, neurodegenerative, and neuropsychiatric disorders which should be further investigated. Similarly, neurological complications that may or may not have been apparent during hospitalization seem to have an impact on the progression of long-COVID. Furthermore, the post-infection long-term neurological complications can also impact young and home-isolated adults with mild COVID-19, independently of high antibody titers after recovery and preexisting diseases complicating the full elucidation of the underlying pathology involved. Considering the organization of all existing knowledge in an AOP framework and incorporating a multiscale approach might further expand our understanding of the mechanisms leading to COVID-19-associated neurological symptoms and help to identify data gaps.

## 5. Conclusions

The current pandemic has offered a unique opportunity to bring together scientists with expertise in various areas including clinicians, in vitro and in vivo experimentalists, computational scientists, and neuroscientists. The CIAO project relies on collaborative, crowdsourced engagement to develop AOPs relevant to COVID-19. The AOPs developed for neurological symptoms following SARS-CoV-2 infection are associated with anosmia, encephalitis, stroke, and seizures. The integration of these AOPs identified neuroinflammation and neurodegeneration as common KEs leading to several of the outcomes. However, besides these putative AOPs, several AOs are still missing or under development to describe other neurological complications observed in COVID-19 patients. Moreover, the understanding of KEs and the overall mechanisms leading to Long-COVID, which is reported to include fatigue, cognitive dysfunction (i.e., brain fog, memory loss, attention deficit), autonomic complications (i.e., orthostatic intolerance, palpitations, and gastrointestinal dysfunction), sleep disturbances, anxiety, and depression are under investigation.

## Figures and Tables

**Figure 1 cells-11-03411-f001:**
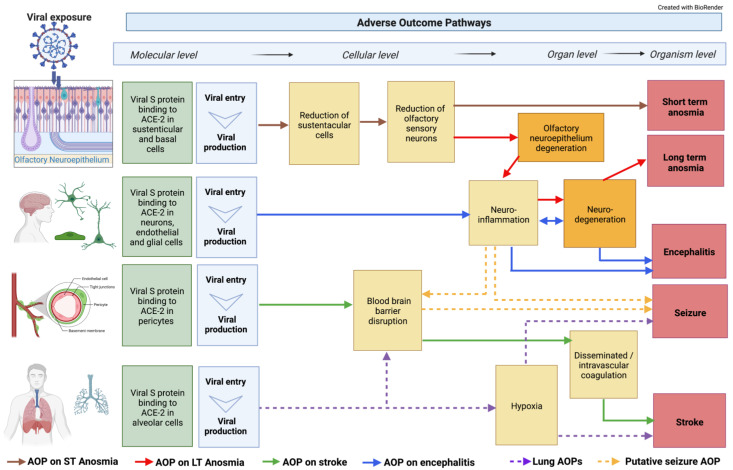
Integration of classical AOPs. Neuronal-related AOPs developed so far within the CIAO dedicated neuro working group integrated into an AOP network. Some of these AOPs are already uploaded on the AOP-Wiki, whereas others are only drafted outside the platform. The AOP network was built on 18 KEs, including 4 MIEs, 8 specialized (neuro-related) Kes, and 4 AOs. Two of these KEs were already available in the AOP-Wiki while the others were developed within the CIAO project. The dotted lines indicate not fully developed AOPs.

**Figure 2 cells-11-03411-f002:**
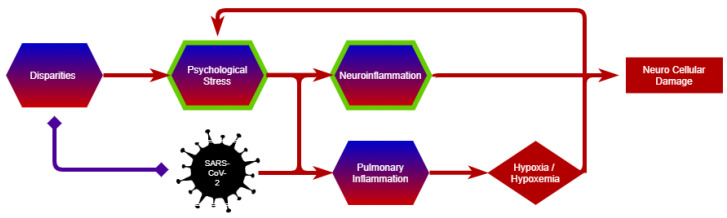
Disparities and sub-pathways to neuro and pulmonary outcomes. The figure illustrates how disparities may instigate psychological stress, leading to both neuroinflammatory and pulmonary inflammation. Pulmonary events can, in turn, lead to hypoxia and additional neurological events. The figure serves as an example of how a multiscale pathway perspective might be presented while concurrently developing the graphical and computational tools that will allow for the representation and analysis of these events and factors across scales.

## Data Availability

Not applicable.

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
