# Peer review of "The Adverse Outcome Pathway Framework Applied to Neurological Symptoms of COVID-19"

_cells, 2022, doi:10.3390/cells11213411_

Round 1

Reviewer 1 Report

1. Please do not use same word keywords and title.
2. Some self-citation in-text requires to justify.
3.The authors need to follow the author's guidelines when they writing the manuscript.
4. Moreover, in the literature, it would be better to add some important papers dedicated to some recent references.
5. Conclusion, the authors must improve the writing, it is not a summary of results.
6. Polish the manuscript grammatically and typically.

Author Response

We are appreciating the reviewers comments and have addressed these in the revised version of the manuscript.

1. Please do not use same word keywords and title.

We have removed the keywords that are used in the title and substituted them with AOP and neuropathology.

2. Some self-citation in-text requires to justify.

We critically reviewed the self-citations but feel some of them are important to keep for the readers to find additional information about the CIAO project for example. In addition, the authors have very few publications together and are identified as experts in various fields, why some of their research is important to include, especially for the AOP framework. Looking at the vast number of references used, a very small proportion is self-citation.

3.The authors need to follow the author's guidelines when they writing the manuscript.

We have changed the format slightly to follow the template better, however we want to highlight that the author guidelines states “Review papers and other article types have a more flexible structure”. 

4. Moreover, in the literature, it would be better to add some important papers dedicated to some recent references. 

We are a bit confused with this comment as majority of the papers are from 2020-2022, more recent seems difficult. If the reviewer has some specific papers in mind, we will happily consider to include them.  

5. Conclusion, the authors must improve the writing, it is not a summary of results.

There was no conclusion in our initial manuscript, we have revised and divided Summary and future directions into two sections Discussion and Conclusion. We hope this is what the reviewer eluate to.  

6. Polish the manuscript grammatically and typically.

We have polished the manuscript.

Reviewer 2 Report

In this study, the authors use the classical AOP approach to investigate the currently available knowledge on neurological manifestations linked to SARS-CoV-2. Overall, the authors provide a reasonably detailed and thorough characterizing neurological AOPs of COVID-19 by modulating multiscale factors such as age, psychological stress, nutrition, poverty, and food insecurity. The text provides good evidence and is worthy of publication. The manuscript is well-written and scientifically sound. It has potentially high interest to readers interested in the area of Neurological Symptoms of COVID-19. I suggest the authors consider the following points as they revise their manuscript and continue their work in this important (and interesting) research area.

My specific comments are mentioned below:

The abstract should be supplemented with specific, most important results, not only a description but specific data and conclusions following them.

Figure 1. All arrow lines should be equal in weight, and the labelled text size should be larger.

The introduction needs minor revision. The author could check a recently published article on COVID-19: Neurological manifestations in COVID-19 patients.

Section five “summary and future directions” Please make it concise, specific and short.

Author Response

We really appreciate the comments of the reviewer that increase the quality of our manuscript.

In this study, the authors use the classical AOP approach to investigate the currently available knowledge on neurological manifestations linked to SARS-CoV-2. Overall, the authors provide a reasonably detailed and thorough characterizing neurological AOPs of COVID-19 by modulating multiscale factors such as age, psychological stress, nutrition, poverty, and food insecurity. The text provides good evidence and is worthy of publication. The manuscript is well-written and scientifically sound. It has potentially high interest to readers interested in the area of Neurological Symptoms of COVID-19. I suggest the authors consider the following points as they revise their manuscript and continue their work in this important (and interesting) research area.  

My specific comments are mentioned below: 

  1. The abstract should be supplemented with specific, most important results, not only a description but specific data and conclusions following them.

We have shortened the abstract to include the most important data and conclusion. It now also follows the template to be about 200 words.

  1. Figure 1. All arrow lines should be equal in weight, and the labelled text size should be larger.

We have updated the figure accordingly.   

  1. The introduction needs minor revision. The author could check a recently published article on COVID-19: Neurological manifestations in COVID-19 patients.

We appreciate the recommendation to include this paper. In addition, we identified some other articles looking at neurological manifestation in patients. Other minor revisions has also been included in the introduction.

  1. Section five “summary and future directions” Please make it concise, specific and short.

We agree with this comment and have divided the summary and future direction into two sections, discussion and a short conclusion. 

Reviewer 3 Report

Submitted manuscript is review article whose aim is to organize the available knowledge on the neurobiological mechanisms underlying corona virus disease 2019 (COVID-19) using the Adverse Outcome Pathway (AOP) framework. This topic is extremely current and the work is based on the most up-to-date data. It is well known that many outcomes observed in COVID-19 are associated with the nervous system; however, the precise mechanisms by which SARS-CoV-2 virus induces neurologic damage are not clear. Appropriate methodology was applied for this type of article, and contemporary literature was cited. After adequate introduction, biological key events were described such as: binding to ACE2, blood brain barrier disruption, hypoxia, neuroinflammation and oxidative stress. Based on collected scientific evidence authors developed four AOPs: anosmia, encephalitis, stroke, and seizure that were associated with neurological adverse outcomes. Construction of AOP networks offered a unique visualization of the core pathways and shared mechanisms, with great practical impact. The authors also identified and underlined some challenges with applying the AOP framework to such a complex disease. For example, the impact on the neurological AOPs of COVID-19 by modulating and multiscale factors such as age, psychological stress, nutrition, poverty, and food insecurity was investigated. It was explored how these factors as well as their impact on the disease could be encompassed and integrated into new multiscale pathway perspectives. Future directions for work in this field have been offered also. In my opinion this manuscript could be extremely useful for both basic scientists and medical practice. I truly recommend it for publication in its present form.      

Author Response

We want to thank the reviewer for this very nice feedback. We hope this will be useful for a broad community and we aim to continue our work.  

Round 2

Reviewer 1 Report

Accept